# Is There a Relationship between Idiopathic Scoliosis and Body Mass? A Scoping Review

**DOI:** 10.3390/nu14194011

**Published:** 2022-09-27

**Authors:** Dalila Scaturro, Agnese Balbo, Fabio Vitagliani, Leonardo Stramazzo, Lawrence Camarda, Giulia Letizia Mauro

**Affiliations:** 1Department of Surgical, Oncological and Stomatological Disciplines, University of Palermo, Via del Vespro, 129, 90127 Palermo, Italy; 2Faculty of Medicine and Surgery, University of Catania, Via Santa Sofia 87, 95100 Catania, Italy

**Keywords:** idiopathic scoliosis, body mass index, weight, body composition, spinal deformity

## Abstract

The etiopathogenesis of idiopathic scoliosis remains unknown, although genetic or hereditary factors, neurological disorders, hormonal and metabolic dysfunctions, biomechanical factors, and environmental factors seem to be involved. Several studies have found that patients with scoliosis have common characteristics of taller stature, lower body mass index (BMI), and low systemic bone mass. We conducted a scoping review to analyze the association between idiopathic scoliosis and BMI. The search for articles was performed on PubMed and Cochrane, including the English language, full-text and free-full-text articles published from 31 December 2011 to 31 December 2021. Most of the results analyzed are in favor of a relationship between low BMI and scoliosis. Having a low BMI could be associated with the presence of scoliosis, although the reason for which is still doubtful. However, further large-scale epidemiological studies on different ethnicities and a comparison of BMI with the healthy population will be needed to better define the correlation between BMI and scoliosis.

## 1. Introduction

Idiopathic scoliosis is a three-dimensional deformity of the spine in the three planes of space, characterized by rotation of the vertebrae and lateral deviation of the spine; radiographically, scoliosis is defined as a Cobb angle greater than 10° on X-ray of the spine, a universally used diagnostic criterion since 1948 [1,2].

It is the most common cause of pediatric spinal deformities. Its prevalence is between 1 and 2% in school age, up to the age of 15, with a higher incidence in women [3,4,5]. Idiopathic scoliosis has no known cause and is divided into 3 groups based on the age of onset: infant (0–3 years), juvenile (4–10 years) and adolescent (AIS) (10–19 years) [1,2]. About 80% of scoliosis cases are AIS, thus defined for the adolescent age range which, according to the WHO, is from 10 to 19 years, with a prevalence of 3% in the general population [6,7].

The multifactorial etiology of scoliosis is now accepted, in which various factors are involved, such as genetic or hereditary, neurological disorders, hormonal and metabolic dysfunctions, biomechanical factors and environmental factors [8]. It is also accepted that scoliosis is a systemic disease and that it mainly results from abnormal systemic skeletal growth and asynchronous spinal neuro-bony growth [9].

Various possible risk factors related to the onset of scoliosis have been studied in the literature, such as growth alterations, postural alterations, heavy backpacks, environmental factors, high-risk sports, visual and dental disturbances and BMI [10,11,12,13]. One of the most questioned and conflicting risk factors is BMI in scoliosis. There are few studies in the literature on this subject and their opinions appear quite conflicting. Notably, several studies had found that patients with scoliosis have common characteristics of taller stature, low systemic bone mass, and lower body mass index (BMI) [14]. Conversely, other authors have highlighted the presence of large scoliotic curves in overweight and obese patients [15]. Still, others have stated that having a high BMI is protective against scoliosis [10].

The identification of risk factors appears to be of fundamental importance during the screening phase, which represents one of the most important secondary preventive tools. It allows for the implementation of adequate prevention strategies, especially in the groups most at risk (i.e., in the case of AIS, the age group 11–14 years) [16,17,18]. In many countries, screening programs have been established to detect deformity early while the spine is still amenable to bracing.

Based on the above, we decided to conduct a scoping review with the aim of analyzing the relationship and/or association between BMI and scoliosis, in order to refer patients most at risk to early screening.

## 2. Materials and Methods

This scoping review was performed according to the Preferred Reporting Items for Systematic Reviews and Meta-Analyzes Extension for Scoping Reviews (PRISMA-ScR) model [11]. The Technical Expert Group (TEP) was made up of 6 physicians, including 4 physiatrists with experience in Idiopathic Scoliosis (G.L.M, D.S., F.V., A.B.) and 2 orthopedic surgeons (L.C., L.S.).

The TEP sought to investigate the effects of BMI as a risk factor for the development of idiopathic scoliosis.

TEP carried out its search on PubMed (Public MedLine, managed by the National Center of Biotechnology Information (NCBI) of the National Library of Medicine in Bethesda (Bethesda, MD, USA), and on Cochrane Library, using the following search string “idiopathic scoliosis mass”, “scoliosis body mass index” “scoliosis weight” “body mass index spinal deformity”, “body composition spinal deformity” and “body composition scoliosis”.

The TEP has taken into consideration, for the purposes of admissibility, the articles published from 31 December 2011 to 31 December 2021.

Two reviewers (F.V. and A.B.) independently reviewed qualifications and abstracts of eligible studies.

Inclusion criteria: articles in English; publication date between 31 December 2011 and 31 December 2021; indexed studies that included assessment of BMI as a risk factor for idiopathic scoliosis; and studies that included patients between 4 and 19 years old. Exclusion criteria: systemic reviews; review; book and documents; meta-analysis; studies that included patients over 19 years old (see Table 1).

The exploration of the heterogeneity of the studies was carried out by evaluating their quality (i.e., the level of evidence) in order to adequately consider the suitability of a scoping review (overall or by a subgroup of studies).

Two reviewers (D.S. and G.L.M.) assessed the risk of bias for the included study using the Cochrane Risk of Bias Assessment Tool available through Covidence. Conflicts were resolved by the Senior Auditor (D.S.). The risk of bias assessment included 13 relevant criteria (see Table 2). Each criterion was rated as presenting a high, low, or unclear risk of bias. A study was judged to be of high quality if all of the criteria were reported to have a low risk of bias.

## 3. Results

We initially found 114 articles from the database of PubMed and 226 articles from the database of Cochrane Library. After the exclusion of 272 articles because of duplicity, language, pertinence and clinical characteristics, we excluded other 47 papers after reading the titles and abstracts. Further, 11 articles were excluded after reading the full text because they did not meet our inclusion criteria. The remaining 10 articles (published between 2011 and 2021) met the inclusion criteria (Figure 1). All studies included in our analysis were focused on the relationship between scoliosis and BMI. Of these studies, 7 showed an association between low BMI and scoliosis, 1 showed no correlation between BMI and scoliosis, and 2 showed an association between high BMI and scoliosis.

### 3.1. Demographic Data

A total of 859,510 patients were evaluated across the studies included in the review. Of these, 10,591 patients already had or were diagnosed (in screening studies or during the studies) with scoliosis. The age of the patient across all of the studies was in the range of 6–19 years (see Table 3).

### 3.2. Body Measurements

The body measurements taken into consideration as a parameter of comparison between the different studies were the BMI and the degree of scoliosis assessed by the Cobb angle. Except for the study of Kyoung-kyu Jeon et al. [19], where scoliosis was assessed using a stadiometer (Seca, Hamburg, Germany) and a body composition analyzer (Inbody 720, Biospace, Seoul, Korea), and the studies of M. Clark et al. [20] and Weijun Wang et al. [21], that used a method of measuring scoliosis from DXA images (the DXA Scoliosis Method, DSM), to reduce the exposition of children to X-radiation, all other studies evaluated the presence and degree of scoliosis using radiographs and Cobb angle measurements (common reference value for diagnosing scoliosis was a Cobb angle ≥10°). There were 3 BMI criteria used in the studies, always calculated with the kg/m^2^ formula, they included: the criteria of the WHO, the International Obesity Task Force and the Korean Centers for Disease Control (KCDC). According to the criteria defined by the WHO, the subject will have severe thinness with a BMI ≤ 16.00, will be underweight with a BMI between 16.00 and 18.49, will be normal weight with a BMI between 18.5 and 24.99, will be overweight with a BMI between 25.00 and 29.99 and obese with a BMI of ≥30.00. The criteria defined by the International Obesity Task Force are based on the fact that the BMI limits they define overweight and obesity in children according to age and sex must be found on the BMI percentile curves that exceed, at the age of 18, the values of 25 kg/m^2^ and 30 kg/m^2^, respectively, for overweight and obesity. Finally, according to the criteria of the Korean Centers for Disease Control (KCDC), obesity was defined as a BMI of >95th percentile, and over-weight was defined as a BMI of >85th percentile and <95th percentile. Normal weight was defined as a BMI of >5th percentile and <85th percentile, and wasting was defined as a BMI of <5th percentile, based on BMI-for-age growth charts (girls/boys aged 2–18 years). 

### 3.3. Body Mass and AIS

#### 3.3.1. Studies That Found a Relationship between Low BMI and Scoliosis

Jeon et al. [18] conducted a 2-year cross-sectional study that enrolled randomly selected subjects from elementary and middle schools in Korea. A total of 1375 Korean children with scoliosis (749 males and 626 females) aged 10.86 ± 1.29 (m) 10.89 ± 1.28 (f) were included in the present study. They were divided into three groups based on their BMI levels (SUW severely underweight: BMI < 16, UW underweight: BMI between 16 and 18.5 and the normal weight group NW: BMI between 18.5 and 25) to study the effect of BMI on risk factors and the onset of idiopathic scoliosis. Participants underwent body composition measurements and screening for idiopathic scoliosis. Children with a BMI above 25 were excluded. Height, weight and body composition were studied using a stadiometer (Seca, Germany) and a body composition analyzer (Inbody 720, Biospace, Korea). X-ray examinations of the spine were not performed but the deformations of the spine were assessed through the use of equipment for the analysis of the structure of the spine (Formetric 4 D) by means of a video rasterstereography (standard error of means within 3°). A covariance analysis with age and gender adjustments was performed to compare the groups based on body composition and spinal deformity condition. Subsequently, a logistic regression analysis was performed with adjustments for age and gender to compare the odds ratios (Ors) of idiopathic scoliosis based on BMI. They found that low body weight was significantly associated with the incidence of scoliosis and that a higher BMI was protective against the development of AIS in both adolescents and women. The angle of scoliosis was significantly greater in the SUW group than in the NW group (SUW: 12.05° ± 4.77°; NW: 10.96° ± 4.54°, *p* < 0.05). Furthermore, boys showed significantly smaller spinal curvature angles than girls (boys: 11.04° ± 4.49°; females: 11.55° ± 4.81°, *p* = 0.041).

Kim et al. [10] conducted an analysis of secondary data on a national health examination carried out annually by the Korean Ministry of Education under the School Health Act. 16412 Korean students were included in the study, of which the prevalence of scoliosis was 2.6%, or 434 children (1.6% men and 3.8% women). Several scoliosis-related parameters were evaluated, including sex, BMI, sleep, and exercise. To analyze the data, the authors used a bivariate analysis, and chi-squared testing was performed to assess associations between individual factors and scoliosis in adolescents. In addition, multiple logistic regression by sex was performed to assess the factors influencing scoliosis, which included variables identified as significant in the bivariate analysis. No relationship with scoliosis was found in the last two parameters. Regarding the correlation between sex and scoliosis, the prevalence of scoliosis was found to be higher in women than in men (*p* < 0.001). BMI was assessed using various criteria, those of IOTF (International Obesity Task Force), WHO and KCDC (Korean Centers for Disease Control). The prevalence of thin students with AIS was found to differ according to the BMI criteria. In men, the prevalence of thinness among adolescents with AIS was 9.0%, 3.8%, and 10.5%, respectively, according to IOTS, WHO, and KCDC. In women, the prevalence of thinness among adolescents with AIS was 14.3%, 3.3%, and 12.0%, respectively, according to IOTF, WHO, and KCDC. All of the criteria were significant among the female students. Therefore, the study proposed greater control on thin adolescents aimed at an early management of AIS.

The study conducted by Miyagi et al. [22] included 210 girls diagnosed with AIS, aged 10 to 18 years. Patients’ body composition was studied by evaluating BMI, Lean Muscle Mass Index, Body Fat Percentage, and Estimated Bone Mass Index. In a second analysis, 111 patients with complete bone maturation were included (Risser 5 or Risser 4 with menarche for more than 2 years) who were divided into two groups (those with Cobb angle <40° and those with Cobb angle> 40°) to study the correlation between the Cobb angle and each body composition parameter. Comparisons of body composition parameters between the two groups were performed using Student’s t-tests. A *p* value < 0.05 was considered significant. The correlations between the Cobb angle and the body composition parameters were instead evaluated using the Pearson correlation coefficients. It was found that BMI was significantly lower in the severe scoliosis group than in patients with moderate scoliosis (*p* < 0.05) and that the lean muscle mass index and estimated bone mass index were significantly lower in the severe scoliosis group compared to patients with moderate scoliosis (*p* < 0.05). The authors suggested that high fat mass and low muscle mass could be related to the progression of scoliosis.

Cai et al. [23] first conducted a cross-sectional study involving 5497 students (2479 females and 3018 males) to examine the morphology and prevalence of scoliosis among primary school students in the city of Chaozhou. Later, the authors performed a case-control study based on previous screening involving 2547 children aimed at identifying risk factors associated with scoliosis and providing clues for the prevention, intervention and treatment of idiopathic scoliosis. Questionnaires were administered to assess demographics, postural habits, scoliosis cognition, scoliosis self-sensation with symptoms, physical condition, sleep time, exercise time, and any other factors. To explore the factors associated with scoliosis, the authors used a logistic regression analysis model. The prevalence of scoliosis was 6.15%, with a total of 338 cases of scoliosis (122 males and 216 females). The study results showed that females had a positivity rate of 2.16 times that of males, that children with myopia were 1.49 times more likely to have scoliosis than children without myopia, that the likelihood of scoliosis increased gradually with decreasing sleep time. In addition, the increase in exercise time has resulted in a gradual reduction in the degree of scoliosis. Regarding body weight, the BMI of each individual was assessed and the BMI of children with scoliosis was lower than in the control group (16.03 ± 3.22 kg/m^2^ vs. 16.91 ± 3.95 kg/m^2^), however, both groups were within the normal range of the student’s body weight and physical health.

The association between BMI and scoliosis was further investigated by M. Clark et al. [20], who conducted a prospective study using the Avon Longitudinal Study of Parents and Children (ALSPAC) to evaluate the association between body composition variables (BMI, body weight, lean mass and fat mass) at the age of 10 and scoliosis at the age of 15. The authors initially calculated the probabilities of exposure to the presumed risk factors in those with and without scoliosis. Subsequently, by means of a logistic regression, they calculated the ORs and the 95% confidence intervals to describe the association between the risk factor and the presence or absence of scoliosis and to identify independent associations between body composition variables and scoliosis. Finally, they performed a sensitivity analysis and a cross-sectional analysis excluding those who at age 9 had scoliosis with a Cobb angle ≥10° and those who did not have a completely straight spine. A total of 5299 15-year-olds were included in the study, of whom 312 (5.9%) had scoliosis. Their results showed a negative association between body mass index and scoliosis, taking into account both fat mass and lean mass. Furthermore, the increase in lean body mass over the years has been associated with a 20% reduction in the risk of scoliosis, while an increase in fat mass reduced the risk of scoliosis by 13%. The fact that both lean and fat mass are linked to an increased risk of scoliosis, according to the authors, could be explained by the hypothesis that combined body weight is involved, not the component parts, perhaps acting by traction or other mechanical mechanisms. 

Hershkovic et al. [24] conducted a cross-sectional prevalence study to examine the prevalence of all spinal deformities in adolescents based on their severity and their possible association with BMI and body height. The study cohort consisted of 829,791 adolescents (470,125 males and 359,666 females) with a mean BMI of 22.04 ± 3.8 kg/m^2^. Data for this study was obtained from a medical database containing records of 17-year-old male and female patients prior to their enrollment into compulsory military service. Information on disability codes associated with spinal deformities was retrieved. All subjects diagnosed with spinal deformities were classified into one of three severity groups: mild, intermediate, and severe. Logistic regression models were used to assess the association between BMI and body height at various degrees of spinal deformity based on severity, applying binary models in which spinal deformities were considered as a dichotomous variable and multinomial models without deformed spinal muscles as a basic category. The authors found an overall prevalence of similar mild, moderate and severe spinal deformities in both sexes, with a prevalence of spinal deformities in each group having greater severity in the underweight group than in the normal, overweight and obese groups in both sexes. The prevalence of severe spinal deformities was 3 times higher in the underweight group than in the overweight and obese groups in males and 6 times higher in females. A BMI below normal was associated with an increased risk of spinal deformities, while a BMI above normal was associated with a reduced risk of spinal deformities in both sexes. The greatest association was found between underweight women and severe spinal deformities (OR 53.247, 95% CI: 52.603–4.05, *p* = < 0.001). The ORs of spinal deformities in all degrees of severity were lower in the obese groups of both sexes. The lowest OR was recorded for severe spinal deformities in female and obese patients (OR 50.253, 95% CI: 50.126–0.509, *p* < 0.001). Finally, when analyzing BMI as a continuous variable, it was observed that each increase in one unit of BMI was associated with a decrease in OR of 10.8% and 11.1%, respectively, for males and females.

Wang et al. [21] conducted a case-control study using dual-energy X-ray absorption (DXA) to understand whether male AIS patients have abnormalities in body and bone composition. The authors recruited male patients with AIS and age- and sex-matched healthy controls. Newly diagnosed male AIS patients who were candidates for corrective spinal surgery and had a Cobb angle in the major curve between 40° and 70° were recruited between 12 and 20 years. Logistic regression was used to compare differences in measured data between AIS and control. In addition, adjusted odds ratios (ORs) and 95% confidence intervals (CI) were calculated by checking for confounding factors to identify abnormalities in body composition and bone status in male AIS patients. Standing height and body weight were measured and BMI calculated. Finally, body composition was analyzed using DXA.47 male patients with AIS were recruited, with a mean age of 15.3 ± 2.2 years. Among these patients, 26 had a major thoracic curve, 7 had a major thoracolumbar or lumbar curve, and 14 a major double curve. The mean Cobb angle of the main curve was 52.7° ± 8.0°. Patients with AIS had significantly lower body weight (63.1 ± 15.8 vs. 52.6 ± 9; *p* <0.001) and BMI (21.3 ± 4.3 vs. 18.5 ± 2.6; *p* < 0.001) compared to those of the controls. Furthermore, patients with AIS also had lean mass (43.7 ± 7.6 vs 36.2 ± 6.8; *p* <0.001) and bone mineral content (1.8 ± 0.3 vs. 1.5 ± 0, 3; *p* < 0.001) significantly lower than controls. In contrast, fat mass was comparable between the two groups (12.8 ± 9.3 vs. 9.6 ± 5.3; *p* = 0.049).

#### 3.3.2. Studies That Found a Relationship between High BMI and Scoliosis

Margalit et al. [25] conducted a retrospective cohort study at a US tertiary referral center. They included 483 patients (420 female and 63 male) aged 10 to 18 years, diagnosed with adolescent idiopathic scoliosis who presented to the orthopedic surgeon for initial evaluation of spinal deformity. The included patients had documentation of age, gender, BMI value, major thoracic major curve measured by the Cobb method on radiographs, and main thoracic angle of trunk rotation (ATR) measured with a traditional scoliometer. No patient had undergone treatment for AIS. Records were reviewed for BMI percentile for age and gender (underweight, ≤4th percentile; normal weight, 5th to 84th percentile; overweight, 85th to 94th percentile; obese, ≥95th percentile), patient characteristics, thoracic scoliometer measurements, and major thoracic curves. We used descriptive statistics for demographic information, variance analysis to evaluate differences in main curves, ATR and age at presentation, chi-squared independence tests to evaluate differences in gender distribution, and logistic regression to evaluate the relationship between ATR and main curves. A total of 23 underweight, 372 normal weight, 52 overweight and 36 obese patients were divided according to their BMI. The odds of presenting with a Cobb curve ≥ 20° on radiography were shown to be 4.90 times higher for obese patients than for normal weight patients. Furthermore, there is no significant difference between the odds of having worse scoliosis in overweight or underweight patients compared to patients of normal weight. Given an ATR value of 5°, the odds of presenting with a 20° curve on X-rays were 48% for underweight patients, 64% for normal weight patients, 71% for overweight patients, and 90% for obese patients. This probably occurs because in obese patients the rotational deformity measured in the Adams forward bending test is less visible as it is hidden by the large body habitus, thus leading to later diagnoses.

Matusik et al. [15] conducted a study on 279 patients with newly diagnosed IS (224 females/55 males), aged 14.21 ± 2.75 years, who were in a Rehabilitation department for the first time. The diagnosis was confirmed by both clinical evaluation and standard posterior anterior radiographic examination in standing spinal column with Cobb angle ≥10°. The goal of their study was to correlate the magnitude of the severity of the scoliotic curve with the anthropometric status of patients with idiopathic scoliosis (IS) based on standard anthropometric measurements and bioelectrical impedance analysis (BIA). Patients were divided into 2 groups: moderate scoliosis (10–39°) and severe scoliosis (>40°). Height, weight, waist and hips were studied for each patient circumference and BMI. One-way analysis of variance (ANOVA) was used to analyze any significant differences between the three anthropometric subgroups, namely underweight, normal weight and overweight. Multivariate regression analysis was performed to identify variables influencing the severity of the curve in each anthropometric subgroup. They found that corrected body height, body weight, and BMI were significantly higher in the severe IS group than in the moderate group and that the severity of the scoliotic curve significantly correlated with the degree of adiposity in the IS patients.

#### 3.3.3. Studies That Found No Relationship between BMI and Scoliosis

A cross-sectional study [26] was conducted with the aim of correlating the extent of the severity of the scoliotic curve with markers of bone turnover (osteocalcin (OC) and amino terminal collagen (NTx) crosslinks with respect to leptin level and nutritional status in girls with AIS 77 girls were included in this study with newly diagnosed scoliosis, aged 14.7 ± 2.17 years, divided into three groups based on the Cobb angle detected on radiography (mild scoliosis 10–19°, moderate scoliosis 20–39°, severe scoliosis ≥ 40°). Corrected height, weight and circumferences of waist and hip were measured and BMI, corrected height Z-score, BMI Z-score were calculated (standard deviation number from the 50th percentile) and the waist-to-height ratio (WHtR) for the whole group. Body composition parameters: fat mass (FAT), lean mass (FFM) and predicted muscle mass (PMM) were determined using a bioelectrical impedance analyzer. Markers of bone turnover (osteocalcin (OC) and amino terminal collagen crosslinks (NTx) and leptin levels were evaluated in serum. Age-adjusted multivariate regression analysis and Tanner stage adjusted for identify variables affecting the severity of the curve A one-way analysis of variance (ANOVA) was performed to analyze any significant differences between the three severity subgroups of the curve. Statistical analysis of the data revealed that bone turnover markers (OC and NTx) and leptin were found to be significantly and independently associated with curve severity in the studied AIS girls. Girls with severe AIS subgroup presented significantly more altered bone metabolism than mild and moderate AIS patients, but there were no significant relationships between curve magnitude and anthropometrical status defined by BMI z-score.

## 4. Discussion

During this review, we tried to analyze the possible relationship and/or association between BMI and idiopathic scoliosis, as the knowledge of the risk factors and clinical characteristics of a specific pathology, such as scoliosis, can be useful during the screening phases and in order to make an early diagnosis.

From the review we conducted a positive correlation between low BMI and the presence of scoliosis would seem to be demonstrated, in fact, of the studies examined, 7 out of 10 studies identified the existence of this association. Clark et al. [20] showed that low body mass index, low body weight, low total body fat mass, and total lean body mass under 10 years of age were associated with the presence of scoliosis at age 15.

Idiopathic scoliosis has a high incidence among children all over the world and its multifactorial etiology is now accepted, although to date it is not fully known [16,23,27,28]. This condition can lead to important changes in the child’s physical appearance with a negative psychological effect on it [26].

In the literature, there is little clarity on the possible risk factors for idiopathic scoliosis, as the results appear quite conflicting [16,29,30].

Several studies have shown a genetic basis that explains the hereditary character of idiopathic scoliosis [31,32]. Some authors speak of a genetic mutation linked to the X chromosome (marker GATA172D05 of X chromosome) [33] thus attempting to justify the clear prevalence of scoliosis in females, although the multigenic character of idiopathic scoliosis has long been known, which presents variable phenotypic expressions [32,34,35].

Early screening for scoliosis in children is critically important for the detection and prevention of spinal deformity; however, these screening programs and preventive measures for idiopathic scoliosis are often lacking or absent [17,28,36]. 

Some studies in school-age children showed that children with scoliosis had a higher presence of low body weight than healthy children [37,38,39].

This is probably because, in predisposed subjects, from the biomechanical point of view, the pushing force of the paravertebral muscles is lost, which is essential to counteract the establishment and evolution of the scoliotic curve [26,40,41].

On the other hand, 2 studies highlighted a greater risk of late diagnosis of advanced stage scoliosis in obese patients, hypothesizing that their body mass may hide the vertebral rotation, making the diagnostic process more difficult. Finally, another study found no association between BMI and scoliosis, concluding that BMI and scoliosis are independent factors.

A further result of our review is that the presence of low body weight would appear to be associated with greater severity of spinal deformities [26], while on the contrary a higher-than-normal body weight would appear to have a protective effect [26].

Despite this, the biological mechanism underlying this association remains unclear to date. Some authors have recognized that low BMI is an organic consequence of scoliosis [42,43], while other authors have described it as a risk factor [44].

Some authors report a high incidence of extreme underweight adolescent women diagnosed with AIS, as scoliosis-induced body deformations can generate more attention to diet and calorie intake in young adolescents to control their bodies as much as possible. Indeed, several cases of anorexia and bulimia have been found in adolescent patients with scoliosis [42,43]. However, it is believed that a low body mass index, lower body fat mass, and lower total lean body mass are present before the onset of clinically evident scoliosis, which is a risk factor. Therefore, some researchers’ idea that scoliosis induces low body weight due to body image problems is now less widely accepted, as anthropometric alterations appear first to appear [44].

Evidence in the literature show that a low body mass index and an altered body composition are factors that favor the onset of scoliosis.

Some studies [34,35,45] have hypothesized that the relationship between low body mass index and scoliosis may be a consequence of hormonal problems [31,32,33,34,35,36,37,38,39,40,41,42,43,44,45,46].

Barrios C. et al. [39] argue that some endocrine factors affecting body composition and growth may be involved in the etiology of idiopathic scoliosis. Indeed, in patients with AIS, higher serum levels of ghrelin and lower serum levels of leptin were often found. These are two hormones involved in the regulation of body weight, energy expenditure, and bone metabolism and can be involved in the development of scoliosis [41]. Burwell et al. [47] hypothesized that lower levels of leptin, which plays an important role in brain growth and development, are related to the onset of asynchronous neuro-bony growth, leading to tension in the nevrasse and spinal growth front. He argues that the low effect of leptin on the CNS in individuals with AIS may explain the different patterns of cerebral cortex thinning seen in patients with scoliosis during adolescence, which may be primary (i.e., pathogenetic) or secondary (i.e., adaptation) to the development of scoliosis [47].

Weight and BMI are important indices to reflect a child’s health condition. Maintaining adequate body weight during childhood could help reduce the risk of worsening scoliosis, especially given the growth spurts and development of secondary sexual characteristics that occur during puberty [19]. The assessment and monitoring of body weight could be of great help in the process of screening children at risk of scoliosis, to implementing targeted preventive interventions. Considering what emerges from our review, we can affirm that during the screening phases it is necessary to pay more attention to underweight children, monitoring them more frequently during growth. This gives us an advantage as it allows us to know the populations most at risk and follow them as it should be.

Our review is not without limitations, primarily due to the low-grade evidence of the studies, such as their retrospective design. Second, studies have highlighted the correlation, rather than causality, between the risks of spinal deformity and idiopathic scoliosis and BMI. Additionally, there is significant heterogeneity between the means used to measure and define scoliosis. Finally, as some studies include secondary data analysis, only limited variables were selected; therefore, there may be a significant variability between included studies in terms of data resulting from the selection.

## 5. Conclusions

In conclusion, there is considerable variability in the literature regarding the link between BMI and scoliosis that prevents a firm conclusion. Most studies support a relationship between low BMI and scoliosis. There is a smaller minority of studies that do not support relationship or an opposite relationship. There is no clear consensus on whether or how low BMI causes or is caused by scoliosis, although there are several hypotheses with varying levels of support. It is important to emphasize the importance of screening in young/adolescent subjects as an early diagnosis is indicative of a more effective therapeutic approach, not only in subjects with low BMI but also in obese subjects, in whom the diagnosis is often late. Therefore, despite the results achieved, no definitive conclusions can be drawn and large-scale epidemiological studies on different ethnicities with an in-depth study of biomarkers are needed in order to better define the correlation between BMI and scoliosis.

## Figures and Tables

**Figure 1 nutrients-14-04011-f001:**
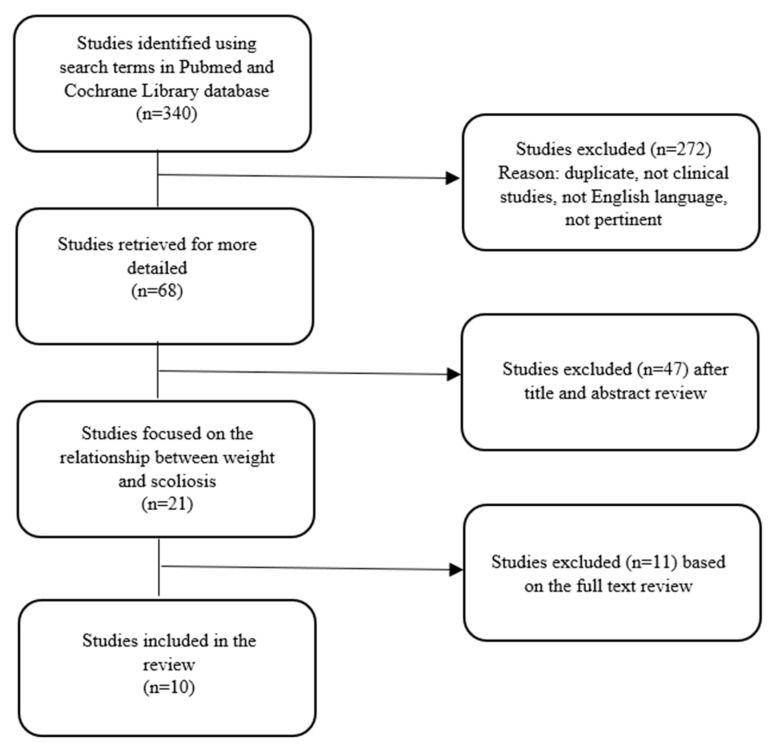
Study selection.

**Table 1 nutrients-14-04011-t001:** Eligibility Criteria.

Inclusion Criteria	Exclusion Criteria
Articles in English Publication date between 31 December 2011 and 31 December 2021 Indexed studies that included assessment of BMI as a risk factor for IS Children with age between 4 and 19 years old	Systematic Reviews Book and documents Review Meta-analysis Studies that included patients over 19 years old

**Table 2 nutrients-14-04011-t002:** Risk of bias assessment.

Criteria	Score
Sequence generation	Low
Allocation concealment	Low
Blinding of participants and personnel	High
Blinding of outcome assessors	High
Incomplete outcome data	Low
Selective outcome reporting	Low
Timing outcome assessments similar?	Unclear
Similarity of baseline characteristics?	Low
Co-intervention avoided or similar?	Low
Compliance acceptable?	Low
Blinding of care provider to the intervention?	High
Randomized participants analyzed in the group to which they were allocated?	Low
Other sources of potential bias?	Low

**Table 3 nutrients-14-04011-t003:** Characteristics (topic, study design, number of patients included or number of studies if revised) and results of studies included in the scoping review on the relationship between body mass and scoliosis.

Author, Year	Study Design	Sample Size	Scoliosis	Weight, BMI	Main Findings
Kyoung-kyu Jeon and Dong-il Kim, 2021	A randomized control study	Total *n* = 1375 Sex: Male 750 and Female 625 Mean age: 10.86 ± 1.29 (m) 10.89 ± 1.28 (f)	Angle of scoliosis: 12.41° ± 4.96° 11.13° ± 4.24° 10.55° ± 4.44° 11.79° ± 4.62° 11.33° ± 5.14° 11.60° ± 4.63°	BOYS BMI < 16 = 113 (SUW) BMI 16–18.5 = 272 (UW) BMI 18.5–25 = 365 (NW) GIRLS BMI < 16 = 157 (SUW) BMI 16–18.5 = 228 (UW) BMI 18.5–25 = 240 (NW)	Scoliosis angle is significantly greater in the SUW group than that in the NW group
Suhee Kim, Ju-Yeon Uhm, Duck-Hee Chae, Yunhee Park, 2020	A cohort retrospective study	Total 16,412 students Sex: Male 8430 and Female 7982 Age: from 7 to 16 years	Scoliosis, n (%): 55 (5.4%) 324 (3.0%) 55 (1.2%)	BMI < 18.5 = n. 1028 (UW) BMI 18.5–25 = n. 10764 (NW) BMI > 25 = n. 4620 (OW)	A higher BMI is protective against the development of AIS. AIS is frequently observed in woman adolescents with a low BMI.
Adam Margalit, Greg McKean, MBChB, Adam Constantine, Carol B. Thompson, R. Jay Lee, MD, and Paul D. Sponseller, 2017	A cohort retrospective study	483 AIS patients Sex: Male 63 and female 420 Mean age: 14 ± 1.6	Main thoracic major curve (°) 33° (22°) 34° (17°) 39° (22°) 44° (17°)	Underweight = 23 Normal weight = 372 Overweight = 52 Obese = 36	Obese patients have 4.9 times higher odds of presenting with a major curve ≥20 degrees compared with normal-weight patients.
Edyta Matusik, Jacek Durmala, Magdalena Olszanecka-Glinianowicz, Jerzy Chudek and Pawel Matusik 2020	A cross-sectional study	77 AIS patients Sex: all female Mean age: 14.7 ± 2.17	mild AIS (10°–19°) moderate AIS (20°–39°) severe AIS (≥ 40°)	BMI 18.13 ± 1.74 = 36 BMI 18.30 ± 2.82 = 30 BMI 19.08 ± 2.30 = 11	The degree of spinal deformity is independently connected with type of the adipose tissue distribution and body composition.
Masayuki Miyagi, Wataru Saito, Takayuki Imura, Toshiyuki Nakazawa, Eiki Shirasawa, Ayumu Kawakubo, Kentaro Uchida, Tsutomu Akazaw, Kazuhide Inage, Seiji Ohtori, Gen Inoueand Masashi Takaso, 2020	A cohort study	210 AIS patients Sex: all female Mean age 14.0 years, range 10–18 years	Moderate scoliosis <40° Severe scoliosis ≥40°	BMI 18.96 ± 2.38 = 87 BMI 18.00 ± 1.96 = 24	The BMI is significantly lower in the severe scoliosis group than that in the moderate scoliosis group (*p* < 0.05).
Zemin Cai, Ruibin Wu, Shukai Zheng, Zhaolong Qiu and Kusheng Wu, 2021	A cross-sectional study and then, a case-control study.	5497 primary school students in Chaozhou city. Then, a case-control study based on the screening involving 2547 children.	AIS group Controls	BMI 16.03 ± 3.22 = 175 BMI 16.91 ± 3.95 = 2372	The BMI of the IS cases is less than that of the controls, but both fall within the normal range of the national standard for students’ physical health.
Emma M. Clark, Hilary J Taylor, Ian Harding, John Hutchinson, Ian Nelson, John E Deanfield, Andy R Ness, Jon H. Tobias 2014	Prospective cohort Study	5299 patients (184 with AIS) Sex: Male 42 and female 142 (with AIS) Age: 15 years	Without scoliosis With scoliosis	BMI 17.6–2.8 BMI 17.2–2.9	There is a negative association between body mass index and scoliosis.
Oded Hershkovich Alon Friedlander, Barak Gordon Harel Arzi, Estela Derazne, Dorit Tzur, Ari Shamiss, Arnon Afek 2013	A cross-sectional study	829,791 patients Sex: 470,125 males and 359,666 females Age: 17 years old	Scoliosis = 7164 59,039 (M) + 44,210 (F)	n. with scoliosis UW = 31,301(M) − 3350 (F) NW = 357,341(M) − 37,395 (F) OW = 48,301(M) − 2647 (F) OBESE = 33,182(M) − 818 (F)	The prevalence of severe spinal deformities is 3 times higher in the underweight group than in the overweight and obese.
Edyta Matusik, Jacek Durmala and Pawel Matusik 2016	A cross-sectional study	279 IS patients Sex: 55 male and 224 female Mean age: 14.21 ± 2.75 years	Moderate:10–39° (*n* = 221) Severe: ≥40° (*n* = 58)	BMI: 18.3 ± 2.85 BMI: 19.19 ± 3.1	The scoliotic curve severity is significantly related to the degree of adiposity in IS patients.
Weijun Wang, Zhiwei Wang, Zezhang Zhu, Feng Zhu & Yong Qiu 2016	A case-control study	87 patients Sex: all male Mean age: 15.3 ± 2.2 years	40 Control 47 AIS patients	BMI: 21.3 ± 4.3 BMI: 18.5 ± 2.6	Patients with AIS have significantly lower body weight and BMI compared to those of the controls.

## Data Availability

Not applicable.

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
