# Peer review of "Is There a Relationship between Idiopathic Scoliosis and Body Mass? A Scoping Review"

_nutrients, 2022, doi:10.3390/nu14194011_

Round 1
Reviewer 1 Report
In the review, Dalila et al. summarize the existing literature on the relationship between BMI and scoliosis. Although no consensus conclusion could be reached by the review, the authors make a comprehensive study on this issue.
Other than merely describing the results of different studies, would the authors provide some evaluation of methods in these studies?
Author Response
Thanks for the tip. We have modified the article as recommended by you.
Reviewer 2 Report
Comments to the Authors of manuscript number: nutrients-1933783 entitled “s there a relationship between idiopathic scoliosis and body mass? A scoping review”.
Authors have presented the review relating to the idiopathic scoliosis and BMI. It is well performed review with many details well described.
1. L 21 – is this problem ethnic?
2. Introduction is well written, consistent, however there is the lack of the aforementioned information that the problem of scoliosis affects various ethnic groups
3. L 65 – the technical expert group is well presented
4. L 70-75 – the keywords and sites where articles were looking and the range of time within article are published is defined and well presented
5. L 86 – a meta-analysis or systemic review?
6. Authors have presented inclusion and exclusion criteria as well as the way how studies were selected. The table and figure are very helpful
7. even the risk of bias assessment is presented
8. BMI and the degree of scoliosis is well described based on the various studies.
9. Table 3- what is: “the SR”. It should be explained. On the other hand table is very informative in many details.
10. L 290 – “on him”?
11. although Authors have presented many studies describing scoliosis in Korean children, there is a lack of discussion about it, and further They concluded that it could be ethnic problem. The ethnic risk should be mentioned in the discussion.
Author Response
- Thanks for the question. This is not an ethnic problem. However, having selected, according to the inclusion and exclusion criteria, studies carried out on a few ethnic groups, we considered ethnicity a limit and probably a reason for encouragement for carrying out further studies.
- Thanks for the comment. As mentioned earlier, we did not take into account the ethnic variable in the populations of the studies under examination.
3) Thanks.
4) Thanks.
5) Thanks. We corrected the wording with the study model used, i.e. a scoping review.
6) Thanks.
7) Thanks for the comment. SR is the acronym for Scoping Review. We have changed the wording to be clearer.
8) Thanks. For this reason, we have stated that we need other studies on the evaluation of the ethnic component in the relationship between BMI and scoliosis, this is our limitation.
Reviewer 3 Report
Comments nutrients 1933783
I would like to thank the editorial board for providing me this opportunity to review this paper.
It is a good review that was written very well. However, I will be addressing few comments that found to be somehow critical to publish the manuscript:
1. Line 154: why the authors removed those who have high BMI? If this applied, thus the authors needs to concentrate their writing and the title on “low BMI” instead of being general and say BMI.
2. Wondering why BMI>25 was excluded? Is there any impact of obesity or overweight on scoliosis?
3. Line 166: please rephrase the sentence.
4. Line 192: what do you mean by progressive scoliosis?
5. In the paragraph 3.3. I suggest to divide this paragraph by subtitles instead of studies
Also, I am seeing that you included the body composition analysis in your paragraph 3.3. I guess this paragraph should be organized by topic and not by studies. Please revisit and re-organize by topic.
6. The first paragraph in the discussion should be the main findings of your work. Please insert these findings
7. I suggest to restructure the discussion also in the same way of the results: by topics. Because in the current mode, it is so confusing.
I ask the authors to take all these comments into consideration to facilitate the approval of the paper.
Author Response
1) Thanks for the comment. In this study by Jeon et al., the authors included patients with BMI up to 25 for unspecified reasons. In any case, we included this study among those in favor of a correlation between low BMI and scoliosis. We did differently for the studies that correlated high BMI and scoliosis or did not show any correlation.
2) Thanks. A relationship between obesity and scoliosis cannot be ruled out, although most studies are in favor of the relationship between low BMI and scoliosis. In any case, we reported the need to pay attention to obese children with scoliosis during the screening phases because, as we have written, fat could hide the slight scoliotic curves and increase the risk of late diagnosis.
3) Thanks. We have rephrased the sentence better
4) Thanks. We have rephrased the sentence better
5) I thank the reviewer for the suggestion, however, having considered BMI and scoliosis as the main topics, it seems more appropriate to divide the paragraph according to the studies that support the different hypotheses.
6) We thank the reviewer. We have changed the order as you suggested, however, the subdivision into paragraphs does not seem a good choice. We hope it will still be to your liking.
Round 2
Reviewer 3 Report
The manuscript shaped well